# Trends and Outcomes of TAVR: An Analysis Using the National Inpatient Sample and Readmissions Database

**DOI:** 10.3390/diseases13050149

**Published:** 2025-05-13

**Authors:** Vivek Joseph Varughese, Vignesh Krishnan Nagesh, Hadrian Hoang-Vu Tran, Olivia Yessin, Harsh Jha, Ashley Mason, Audrey Thu, Simcha Weissman, Adam Atoot

**Affiliations:** 1Department of Internal Medicine, University of South Carolina, Prisma Health, Columbia, SC 29201, USA; vivekjvarughese@gmail.com (V.J.V.); vgneshkrishnan@gmail.com (V.K.N.); olivia.yessin@prismahealth.org (O.Y.); harsh.jha@prismahealth.org (H.J.); ashley.mason@prismahealth.org (A.M.); 2Department of Internal Medicine, Hackensack Palisades Medical Center, North Bergen, NJ 07047, USA; simcha.weissman@hmhn.org (S.W.); adam.atoot@hmhn.org (A.A.); 3Touro College of Medicine, New York, NY 10027, USA; sphyo@student.touro.edu

**Keywords:** TAVR, aortic stenosis, national inpatient sample, national readmission database, stroke, heart failure

## Abstract

**Background**: Transcatheter aortic valve replacement (TAVR) has become the preferred treatment for severe aortic stenosis in high- and intermediate-risk patients, with expanding indications for lower-risk populations. However, post-procedural complications, such as stroke, conduction disturbances, and heart failure readmissions, remain concerns. The aim of our study is to analyze the national trends in TAVR procedures, in-hospital outcomes, major readmission causes, and the association of risk factors for readmissions following TAVR. **Methods**: We analyzed NIS data (2018–2022) to assess TAVR utilization trends, patient demographics, and in-hospital outcomes. The NRD (2021–2022) was used to evaluate 60-day readmission rates for stroke, complete heart block, and heart failure. Multivariate regression models were employed to identify risk factors having significant association with major readmission causes. **Results**: TAVR utilization increased from 10,788 cases in 2018 to 17,784 in 2022, with a concurrent decrease in in-hospital mortality (1.33% to 0.90%) and length of stay (3.88 to 2.97 days). Of 123,376 TAVR index admissions in 2021, 28,654 patients had 66,100 readmission events (53.57%) in the 60 days following discharge. Heart failure (17,566 cases, 26.57% of readmissions) was the most common readmission cause, followed by complete heart block (1760 cases, 2.66% of readmissions) and stroke (284 cases, 0.42% of readmissions). Predictors of post-TAVR stroke included uncontrolled hypertension (OR 2.29, *p* < 0.001) and chronic heart failure (OR 2.73, *p* < 0.001). Left bundle branch block (LBBB) was strongly associated with complete heart block (OR 12.89, *p* < 0.001) and heart failure readmissions (OR 7.65, *p* < 0.001). **Conclusions**: TAVR utilization has increased with improving perioperative outcomes, but post-TAVR readmissions remain significant, particularly for heart failure, stroke, and conduction disturbances. Pre-procedural uncontrolled hypertension, hyperlipidemia, congestive heart failure, and atrial fibrillation were risk factors with significant association with stroke in the 60 days following TAVR. The presence of documented pre-procedural LBB, RBB, as well as BFB were risk factors with significant association with complete heart block following TAVR placements. Pre-procedural LBB, RBB, BFB, and atrial fibrillation were risk factors having significant association with heart failure readmissions in the 60 days following TAVR.

## 1. Introduction

Transcatheter aortic valve replacement (TAVR) has revolutionized the management of severe aortic stenosis, especially in higher-risk populations [1]. Over the past decade, TAVR has become a more acceptable alternative to surgical aortic valve replacement (SAVR), with studies showing improved outcomes and decreasing perioperative risks [2]. Since its approval, TAVR has been increasingly utilized in intermediate-risk and low-risk patients [3]. Advances in technology, procedural techniques, and patient selection criteria have led to improved outcomes, including lower perioperative mortality and shorter hospital stays [1]. However, despite these advancements, post-procedural complications, such as stroke, conduction disturbances, and recurrent heart failure, remain areas of concern [4].

Understanding trends in TAVR utilization, patient demographics, and post-procedural outcomes is essential for improved patient selection and management. Prior studies have demonstrated a global increase in TAVR procedures, increasing clinician confidence in its safety [5]. The shift toward TAVR in lower-risk populations is driven by evidence of improved outcomes of TAVR in terms of mortality, stroke, and quality of life improvements [6]. However, concerns regarding long-term durability, conduction disturbances, and risk of stroke, especially in patients with atrial fibrillation or other cardiovascular comorbidities, remain [7].

Among post-TAVR complications, embolic stroke is a major concern, with the highest risk occurring within the first 30 days post-procedure [8]. The introduction of devices such as the Sentinel system was intended to mitigate this risk by preventing debris embolization during valve deployment [9]. Identifying patients at the highest risk for post-TAVR stroke remains an important clinical challenge. Conduction disturbances and the need for permanent pacemaker (PPM) implantation following TAVR have also been widely studied [10]. New onset left bundle branch block (LBBB) and right bundle branch block (RBBB) are major predictors of complete heart block requiring pacemaker placement [10]. The PARTNER trials identified RBBB as a significant risk factor for post-TAVR conduction abnormalities, while the predictive value of LBBB is still being explored [11]. The increasing use of TAVR in younger and lower-risk populations is crucial for improving long-term outcomes by reducing the incidence of conduction disturbances.

Heart failure readmissions post-TAVR is another area of interest [12]. Despite improvements in procedural safety, a study suggested that nearly 13% of patients experience heart failure-related hospitalizations within a 1-year follow-up period. Factors such as pre-existing left ventricular dysfunction, atrial fibrillation, and persistent valvular dysfunction contribute to the risk of post-TAVR heart failure exacerbations [12]. Identifying modifiable risk factors and optimizing perioperative management strategies may help reduce readmission rates and improve patient outcomes.

This study utilizes the National Inpatient Sample (NIS) and National Readmissions Database (NRD) to analyze trends in TAVR utilization and in hospital outcomes for the years 2018–2022. Using the National Readmissions Database (NRD) for the years 2021–2022, we are analyzing the major readmission causes in the 60 days following TAVR and patient risk factors holding significant associations with these.

## 2. Methods

NIS for the years 2018–2022 was used for the analysis. STATA 18.5 MP was used for statistical analysis. ICD (International Classification of Diseases) PCS 10 code 02RF3KZ was used to select admissions that underwent TAVR procedure. Age > 18 was used as the inclusion criteria. The total number of admissions requiring TAVR and population trends over the years were analyzed for disparities. The National Readmission Database for the year 2021 and 2022 was used to select readmissions in the 60 days following TAVR procedures. Embolic stroke, complete heart block, and heart failure readmissions were studied in particular. Homogeneity of distribution of risk factors were assessed. Multivariate regression analysis using probit models was used (because of the large sample size) to assess the association of patient factors that held significant association with readmissions for stroke, complete heart block, and heart failure. The perioperative use of the embolic stroke protection system and effect on stroke readmission following TAVR were analyzed. Age, sex, race, and APDRG risk severity index were used as confounders in the regression analysis. Variance of population factors in the index admissions for TAVR, as well as readmissions for stroke, heart block, and heart failure, were analyzed. One-way ANOVA was used for normal distributions, and Kruskal–Wallis test was used for others. Patient factors that had significant variance among the readmission group were selected for the regression analysis along with known risk factors for specific readmission conditions. A two-tailed *p*-value < 0.05 was used to determine statistical significance.

## 3. Results

Using the National Inpatient Sample from the years 2018 to 2022, we identified national trends in TAVR placement using the appropriate PCS 10 code. Socio-demographic stratification, healthcare resource burden, and hospital outcomes are summarized in Table 1.

An increasing trend in the total number of TAVR procedures was seen between 2018 and 2022, with 17,784 procedures in 2022 compared to 10,788 procedures in 2018. The mean age of the admissions undergoing TAVR remained between 77 and 80, and no significant variance across the years was observed in the one-way ANOVA. No significant disparities in terms of median household income were observed in the trends for TAVR procedures over the years. A general declining trend in the mean length of hospital stay, as well as in all-cause mortality during the hospital stay, was observed; however, no variance was observed in the one-way ANOVA across the years. Total hospital charges associated with the admissions for TAVR were observed to have an increasing trend, but no variance was observed across the years. These results are summarized in Figure 1.

## 4. Day Readmission Analysis After TAVR Admissions

We identified 123,376 index admissions for TAVR procedures between the months of January 2021 and October 2022 using the National Readmissions Databases for the years. The index admission events were followed over a period of 60 days to analyze readmission events. From this patient population pool, 28,654 patients had readmission events in the 60-day follow-up period, and a total of 66,100 readmission events were recorded in these 28,654 patients in the 60 days following TAVR.

Major causes of readmissions in the 60 days following discharge were as follows. Results are depicted in Figure 2.
Heart failure: 17,566 events (26.57%);Complete heart block: 1760 events (2.66%);Stroke: 284 events (0.42%);Acute kidney injury: 1079 events (1.63%);Sepsis: 2256 events (3.41%).

### 4.1. Stroke Following TAVR

A total of 284 readmission events following stroke were recorded in the 60 days following discharge for TAVR. This was 0.4% of all the TAVR procedures and 4.1% of all the readmission events in the 60 days following TAVR.

Comorbid factors documented in the index admission population that held significant association with stroke in the 60 days post-TAVR are as follows. These are depicted in a forest plot in Figure 3.
Uncontrolled hypertension: OR 2.29 (1.72–3.15), *p*-value: 0.000;Atrial fibrillation: OR 1.66 (1.30–2.14), *p*-value 0.000;Chronic congestive heart failure: OR 2.73 (2.1–7.3), *p*: 0.000;Uncontrolled hyperlipidemia: OR 2.45 (1.90–3.16), *p*-value: 0.000.

In patients with readmissions for stroke in the 60 days following TAVR, we analyzed the presence of the SENTINEL embolization protection system used during the TAVR procedure. Of the 284 stroke readmissions, 87.8% of patients who had stroke readmissions in the 60 days following TAVR did not have a sentinel embolic protection device during the procedure. A total of 3456 index admissions for TAVR had a documented SENTINEL embolization protection system in place during the procedure. A total of 16 readmission events for stroke were recorded in the 60 days following TAVR among index admissions with documented SENTINEL embolization protective systems used during TAVR. The results are summarized in Figure 4.

### 4.2. Complete Heart Block Following TAVR

Complete heart block was a common complication identified in the 60 days following discharge for TAVR admissions. We stratified admissions for complete heart block after TAVR and analyzed the associations with patient risk factors. We identified 1760 readmission events for complete heart block.

We analyzed the prevalence of LBB (left bundle branch block), RBB (right bundle branch block), and BFB (bifascicular block) in index TAVR admissions that had complete heart block readmissions in the 60 days following TAVR.
LBB: 21.83% (18.06–21.14);RBB: 17.36% (13.96–21.40);Fascicular block: 6.69% (4.62–9.60).

Documented patient factors that held significant association with readmissions required for complete heart block (after regression for age, sex, hypertension, diabetes, hyperlipidemia, atrial fibrillation, and heart failure) were as follows. The results are depicted in Figure 5.
LBB: OR 12.89 (11.36–14.63), *p*-value: 0.008;BFB: OR: 9.75 (7.95–11.97), *p*-value: 0.019;RBB: OR: 9.13 (7.96–10.49), *p*-value: 0.006.

### 4.3. Heart Failure Readmissions Following TAVR

A total of 17,566 readmission events of heart failure were noted in the 60 days following TAVR placements.

Patient factors documented in the index TAVR admissions that held significant association with heart failure readmissions after TAVR (after regressing for age, sex, income, atrial fibrillation, hyperlipidemia, and diabetes) were as follows. Results are summarized in Figure 6.
LBB: OR: 7.65 (7.32–8.00), *p*-value: 0.000;BFB: OR: 5.34 (4.89–5.82), *p*-value: 0.000;RBB: OR: 2.63 (2.45–2.82), *p*-value: 0.000;Atrial fibrillation: 1.52 (1.47–1.67), *p*-value: 0.000.

## 5. Discussion

As per the 2020 AHA/ACC guidelines on the management of valvular heart diseases, a class I recommendation for surgical aortic valve replacement (SAVR) or transcatheter aortic valve replacement (TAVR) is made for symptomatic severe aortic stenosis (AS) with a mean transaortic pressure gradient > 40 mm Hg or transaortic velocity > 4 m/s. In patients with discordant aortic valve areas and pressure gradients, assessment of classic low-flow, low-gradient AS versus paradoxical low-flow low-gradient AS is determined based on the Stroke Volume Index (SVI) and ejection fraction (EF). An SVI < 35 and an EF < 50% will be included under the classic low-flow, low-gradient phenotype, and a dobutamine stress echocardiogram is carried out to determine whether the mean gradient increases or the aortic valve area increases. In paradoxical low-flow, low-gradient AS, aortic calcium score would be utilized to determine whether the patients would be candidates for SAVR or TAVR. In general, the guidelines recommend TAVR for patients > 80 years of age, and for patients between 65 and 80, shared decision-making is carried out for TAVR versus SAVR. In high-risk surgical candidates with an average quality of life expectancy > 1 year, there is a class I recommendation for TAVR [13].

While the number of patients hospitalized with severe aortic stenosis has remained relatively stable during the period from 2018 to 2022, the proportion of patients admitted with severe aortic stenosis undergoing TAVR had a general increasing trend. While in 2018, approximately 12.1% of all patients admitted for aortic stenosis underwent TAVR, this proportion increased to 19.6% of all patients admitted with aortic stenosis by 2022. The total number of TAVRs also had a general increasing trend, with 17,784 TAVRs performed in 2022 compared to 10,788 in 2018. This was a 64.87% increase in 5 years. More liberal recommendations exist for TAVR compared to its introduction in 2012, which is reflected in the increasing trend in TAVR placements. The introduction of TAVR in 2011 was initially associated with higher rates of procedural complications and treatment failure. Adverse outcomes were mainly attributed to the limitations of the first-generation device technology, which required 18F to 24F catheter arterial sheaths, frequent use of transapical and direct aortic access, near-universal use of general anesthesia, and limited operator experience. Over the past decade, however, there has been a transformation in TAVR technology and periprocedural care and rapid evolution of procedural techniques, leading to marked improvements in procedural success and safety. In addition, with the completion of multiple randomized clinical trials, the use of TAVR has expanded to include intermediate-risk and low-risk patients. As a result of these concurrent trends, there have been marked reductions in the rate of most TAVR complications, as well as in procedure-related mortality. Between 2012 and 2018, 30-day mortality decreased from 7% to 2%, the incidence of 30-day composite adverse events decreased from 26% to 11%, and 1-year mortality decreased from 24% to 12% [14]. This finding mirrors trends seen in numerous other national registry studies that similarly find that the volume of TAVR procedures has steadily increased in recent years [15,16,17,18,19], in some cases with a concomitant decrease in the number of patients undergoing surgical valve replacement. Analyzing the total number of admissions for severe AS (Table 1), a drop in total admissions were seen in the 2020–2021 range. This could be attributed to the COVID pandemic. However, this did not affect the total number of TAVR placements during this time period.

In addition, evaluation of patients’ surgical risk profiles, in this case using the AP DRG mortality index category, indicates that TAVR is increasingly being offered to patients with fewer comorbidities. APDRG risk severity index is a variable specified in the NIS that considers the age and comorbidities of the patient and predicts the mortality chances during the hospital stay. The proportion of patients categorized using the AP DRG score as having a minor likelihood of dying while undergoing TAVR has notably expanded, with 19.74% of all TAVR patients in 2022 assigned to this group compared with just 7.43% in 2018. Meanwhile, the percentage of patients in the extreme likelihood of dying category remained constant over the years (Table 1). These findings mirror those of several other large studies which stratified patients using either Society of Thoracic Surgeons (STS) risk score or Euro Score, which similarly find that TAVR is increasingly being offered to lower-risk populations [20,21]. This likely reflects the growing acceptance of TAVR as a safe and efficacious alternative to SAVR, which has resulted in continued expansion of the approved indications to encompass lower-risk patient populations.

The length of hospital stay following TAVR had a declining trend over the study period, with mean hospital stay decreasing from 3.88 days in 2018 to 2.97 days in 2022. This finding is in line with similar database studies analyzing the length of stay among TAVR admissions. The mean of total hospital charges had an increasing trend, increasing linearly from a total cost of USD 195,643.80 in 2018 to a total of USDD 229,794.60 in 2022. However, using one-way ANOVA, the mean length of hospital stay and mean of total charges did not vary between the years (*p*-value of 0.784 and 0.821, respectively). All-cause, in-hospital mortality related to TAVR procedures remained under 1.5% of total procedures performed, with no significant variance across the years (one-way ANOVA, *p*-value: 0.823).

Stroke is a well-recognized life-altering complication of TAVR procedures, with the highest risk being in the first 2 years after the procedure. Strokes that occur early after TAVR (within 30 days) are most likely due to debris embolization, which is why Sentinel protection systems were employed to begin with [22]. Sentinel was not used in 87.8% of patients in our study who had a stroke following TAVR in the first 60 days. Shekhar et al. evaluated the effects of Sentinel use in TAVR procedures using a retrospective study and found that while Sentinel use did not reduce the overall risk of stroke, it reduced the risk of major cerebrovascular accidents. Many patients who undergo TAVR have multiple other comorbidities and are older, which predisposes them to strokes regardless of this procedure. One of the most significant risk factors for stroke in this group of patients is a history of atrial fibrillation. There was a statistically significant association between atrial fibrillation and stroke within the first 60 days of TAVR discharge in our patients. Uncontrolled hypertension, uncontrolled hyperlipidemia, and heart failure were other risk factors documented in the index admission population that held significant association with stroke in the 60 days following TAVR (Figure 3). Okuno et al. [23] used the Swiss TAVI Registry to evaluate short- and long-term predictors of stroke after TAVR. They found that about one-third of patients in that database who had a stroke after TAVR had atrial fibrillation. It is evident that traditional risk factors for CVAs are the same risk factors that are significant in long-term (>30 days post-TAVR) strokes, as reaffirmed by findings in a review written by Mastoris et al. [24]. These more traditional risk factors for stroke are treatable if identified and appropriately followed up.

Analyzing the prevalence of various pre-existing conduction system defects in patients undergoing TAVR who were found to have a complete heart (CHB) in the sixty days following TAVR, left bundle branch block (LBBB), right bundle branch block (RBB), and bifascicular block (BFB) were found to have significant association (Figure 5). A review in the *Journal of the American College of Cardiology* (JACC) states that there is conflicting evidence to suggest whether LBBB is a predictor of permanent pacemaker (PPM) implantation for complete heart block following TAVR [25]. An analysis of the PARTNER trial in the JACC interventional journal did not list LBBB as a predictor of PPM implantation [26]. The same trial did, however, list RBBB as a risk factor for needing a PPM within 30 days post-TAVR. The Emory risk score used to predict the need for PPM after TAVR does not list LBBB or BFB as risk factors [27]; however, our data showed that both were associated with OR around 9. One of the rationales for not including LBBB in risk stratification protocols for PPM implantation is that about 50% of those with new-onset, post-TAVR LBBB will have normalization of their EKG findings [28]. Our data may provide reasons to reconsider this stance.

Heart failure (HF) remained the major cause of readmission in the sixty days following TAVR. Documented LBB, BFB, RBB, and atrial fibrillation in the index admissions for TAVR were factors that held significant association with heart failure readmission (Figure 6). Previous studies have shown readmissions rates in the one year following TAVR ranging from 25.4% to 52.2% and approximately half of these readmissions being of cardiovascular causes. HF readmissions can significantly impact the long-term outcomes of TAVR based on these study results [29]. In a retrospective analysis of one-year outcomes following TAVR, previous myocardial infarction, diabetes, atrial fibrillation, CKD, and pulmonary hypertension were found to be significant risk factors predictive of heart failure readmissions. It was also noted in this study that the use of angiotensin-converting enzyme inhibitor/aldosterone receptor therapy at >50% of the optimal dose had a significant impact in preventing HF readmissions.

## 6. Conclusions

TAVR utilization had a general increasing trend, with a 64.87% increase in total TAVR admissions in 2022 compared to 2018. All-cause mortality remained less than 1.5% across the years. The length of hospital stay had a general declining trend, while total charges had a general increasing trend, although no statistically significant variance was observed for these variables over the years. Pre-procedural uncontrolled hypertension, hyperlipidemia, congestive heart failure, and atrial fibrillation were risk factors with significant association with stroke in the 60 days following TAVR. SENTINEL embolization protection systems had a significant protective effect in preventing stroke readmissions in the 60 days following TAVR. The presence of documented pre-procedural LBB, RBB, as well as BFB were risk factors with significant association with complete heart block following TAVR placements. Pre-procedural LBB, RBB, BFB, and atrial fibrillation were risk factors having significant association with heart failure readmissions in the 60 days following TAVR.

### Limitations of the Study

Although NIS and NRD are the largest publicly available databases, the issue of selection bias remains a limitation. The patient selection is based on ICD 10, as well as PCS codes; hence, inter-operator variability cannot be accounted for. However, variance in coding for procedures like TAVR is minimal to none, hence, not a limitation in selecting the index population. Although NRD allowed the study of the sample in a longitudinal manner, the diagnosis of LBB, RBB, and BFB in the index admission group had the disadvantage of not being able to be monitored longitudinally.

## Figures and Tables

**Figure 1 diseases-13-00149-f001:**
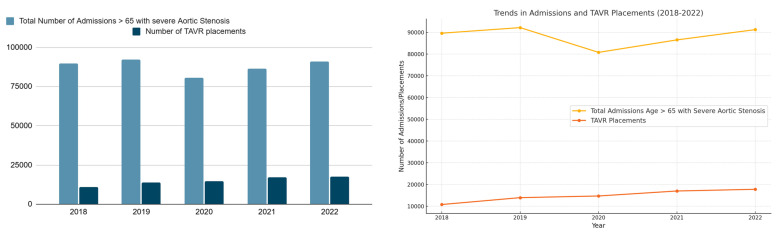
Trends in TAVR placement (2018–2022).

**Figure 2 diseases-13-00149-f002:**
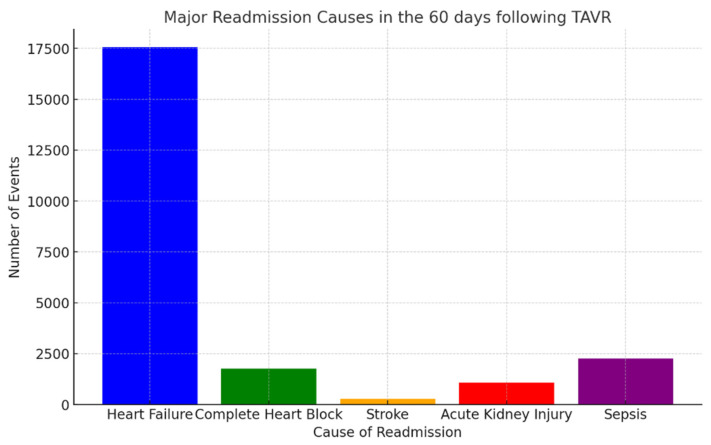
Major readmission causes in the 60 days post-TAVR.

**Figure 3 diseases-13-00149-f003:**
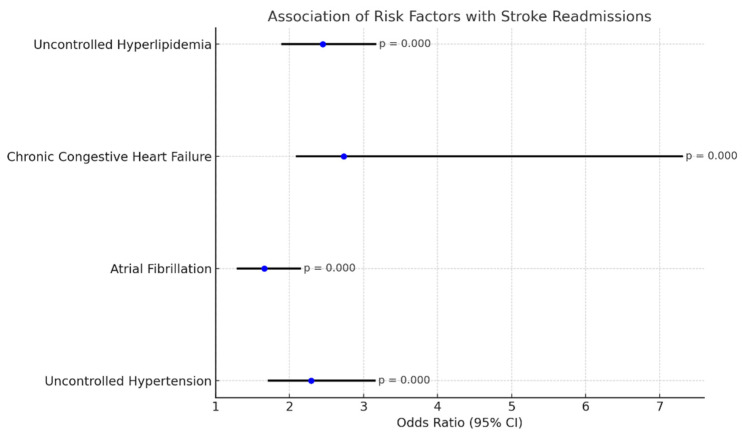
Forest plot for association of risk factors with stroke readmissions.

**Figure 4 diseases-13-00149-f004:**
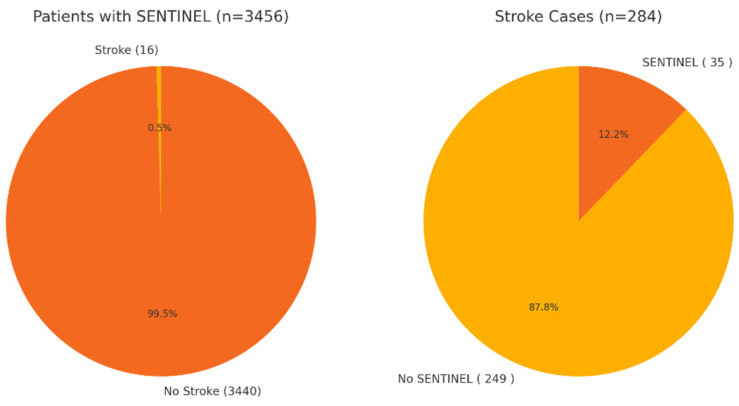
SENTINEL device use and effect on stroke readmissions following TAVR.

**Figure 5 diseases-13-00149-f005:**
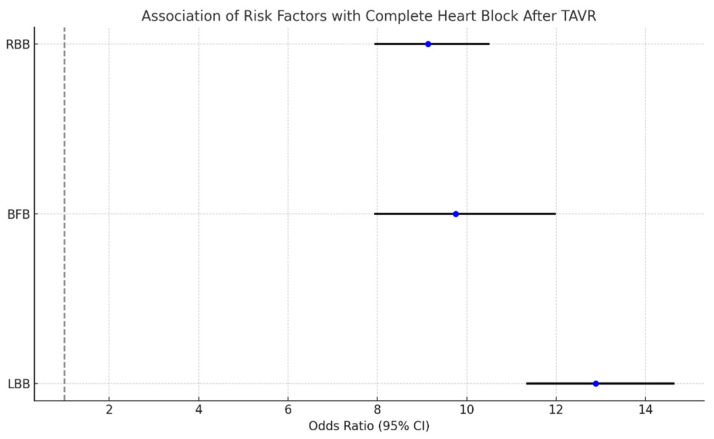
Forest plot for the association of risk factors with readmissions for complete heart block following TAVR.

**Figure 6 diseases-13-00149-f006:**
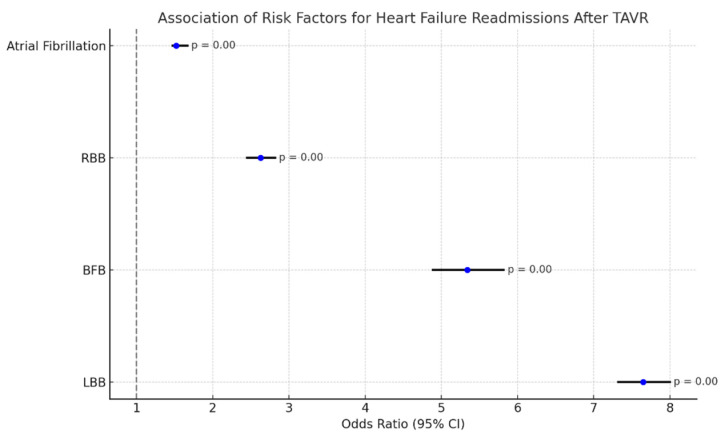
Forest plot for the association of risk factors with readmissions for heart failure readmissions following TAVR.

**Table 1 diseases-13-00149-t001:** Socioeconomic and healthcare resource burden trends for TAVR procedures (2018–2022).

		2018	2019	2020	2021	2022
Total number of admissions with aortic stenosis (Age > 65)		89,548	92,111	80,727	86,466	91,190
Total number of TAVR placements		10,788	13,944	14,708	17,001	17,784
Mean age		79.18 (79.02–79.34)	78.47 (78.32–78.61)	77.74 (77.60–77.88)	77.91 (77.79–78.04)	78.04 (77.92–78.16)
Sex (%)	Male:	53.50 (5717/10,788)	56.03 (7808/13,944)	58.14 (8530/14,708)	57.85 (9690/17,001)	57.05 (10,136/17,784)
	Female:	46.50 (4962/10,788)	43.97 (5995/13,944)	41.86 (6030/14,708)	42.15 (7140/17,001)	42.95 (7469/17,784)
Mean hospital length of stay (days)		3.88 (3.78–3.98)	3.43 (3.35–3.51)	3.22 (3.14–3.31)	3.15 (3.07–3.25)	2.97 (2.90–3.050)
Mean of total hospital charges (USD)		19,5643.8	20,6781.4	219,738.0	224,105.6	229,794.6
In-hospital mortality (%)		1.33% (1.134–1.596%) (143/10,788)	1.15% (0.98–1.34%) (160/13,944)	1.16% (1.07–1.35%) (170/14,708)	1.03% (0.893–1.19%) (175/17,001)	0.90% (0.776–1.05%) (160/17,784)
Race (%)	White	86.83 (9277)	87.59 (12,131)	87.68 (12,795)	87.80 (14,790)	87.11 (15,472)
	Black	4.16 (448)	4.06 (566)	3.96 (582)	3.81 (647)	3.92 (697)
	Hispanic	5.44 (586)	4.43 (617)	4.74 (697)	4.43 (731)	5.25 (933)
	Others	1.38	1.25	1.40	1.49	1.52
AP DRG Mortality Index category (%)	Minor likelihood of dying	7.43	15.18	16.28	18.41	19.74
	Moderate likelihood of dying	44.61	48.31	47.51	48.71	48.89
	Major likelihood of dying	37.97	28.28	32.34	25.26	24.21
	Extreme likelihood of dying	9.98	8.23	10.23	7.62	7.15

## Data Availability

Research done using National Inpatient Sample/National Readmission Database. These are publicly available database in the HCUP website.

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
