# Peer review of "Trends and Outcomes of TAVR: An Analysis Using the National Inpatient Sample and Readmissions Database"

_diseases, 2025, doi:10.3390/diseases13050149_

Round 1
Reviewer 1 Report
Comments and Suggestions for Authors
The authors presented the analysis of the large population of patients subjected to TAVR using the largest publicly available information. These data will be interesting to readers. There is one moment that should be taken into consideration to improve the logic of presentation. The authors postulate the following aim of the manuscript which is “to improve patient selection, procedural techniques, and post-discharge management strategies for optimizing TAVR outcomes”. This aim is very far from what has been done. In reality, the authors have done two major things which are as follows:
- Presented the dynamics of annual number of TAVRs
- Identified the risk factors for TAVR re-admissions
This should be clearly reflected in the goals of the study as well as in the Conclusions. Conclusions should be included not only in the Abstract but also to the main text.
Author Response
The authors presented the analysis of the large population of patients subjected to TAVR using the largest publicly available information. This data will be interesting to readers. There is one moment that should be taken into consideration to improve the logic of presentation. The authors postulate the following aim of the manuscript which is “to improve patient selection, procedural techniques, and post-discharge management strategies for optimizing TAVR outcomes”. This aim is very far from what has been done. In reality, the authors have done two major things which are as follows:
- Presented the dynamics of annual number of TAVRs
- Identified the risk factors for TAVR re-admissions
This should be clearly reflected in the goals of the study as well as in the Conclusions. Conclusions should be included not only in the Abstract but also to the main text.
REPLY: the aim of the study has been changes to trends in the total number of TAVR placements and identification of risk factors associated with readmissions following TAVR placements. Conclusion section has been added to the manuscript
Reviewer 2 Report
Comments and Suggestions for Authors
Thanks you for asking me to review this manuscript
This study evaluates the NIS and NRD databases over 4 yrs for TAVR trends and results.
Subject is relevant in the context of increasing TAVR numbers worldwide and increasing uptake for intermediate and even low risk patients. NID data was used from 2018-2022. It is not clear why the NRD re-admission data was for the years 2021-2022 only (line 84)?
They reported increased TAVR utilization, decreased mortality but high readmissions after TAVR. The data for readmissions was not analysed for TAVR risk group which would have been more meaningful. The manuscript would benefit from some re-organization and better meaningful presentation of data. The presentation of data is unclear and tables have not been adequately utilized for a clearer presentation. Some of the figures are irrelevant and not needed for the data already presented in the text. The results as they are presented are very incoherent and it is difficult to comprehend or summarize from a read. The authors have not written a conclusion statement to their study in the main body of the text. Please get a statistician to review the manuscript before submission.
Please consider the following.
In the abstract, please present data as percentages rather than absolute case numbers.
Please define all abbreviations at first use – STATA, PCS code etc
Line 22 - …..28,654 patients had 66,100 readmission events… - please quote this readmissions /patient over the 60 day period.
Line 22-23 – please give percentages for heart failure, heart block and stroke.
Tables – give percentages for all numbers as appropriate
Provide p values for group comparisons for the years.
Predictors of re-admission have not been analysed.
Line 91 – please describe the APDRG risk severity index and provide reference
Line 104 – please remove this figure as it does not convey much other than what is presented in results.
Table 2 – This refers to socioeconomic data but there is none of that here. It has age, gender and LOS. There is no data on race or income here that would constitute socioeconomic data. Please check the numbers for Total Hospital Charges ($) 72427.6 and 72427.6. They are the same for both index and readmissions.
Line 117 – 123; present this data as readmissions /patient and give overall percentages for each.
Line 135 -138; are the numbers with confidence presented ORs or correlation co-efficients?
Please present this as tables for all (stroke, heart failure and heart block)
Line 133 – says that AF was no significant. This is inconsistent with what is reported in line 136 - Atrial fibrillation: 1.66(1.30 - 2.14) p value 0.000
Line 144 - ……..prevalence of LBB, RBB and BFB in TAVR admissions that had complete heart block readmissions…… Were the LBB, RBB, BFB at the time of discharge since the readmissions were clearly for CHB. Abbreviations not described.
Line 187 – 189; LOS declined substantially - how much is substantial? Is the change statistically significant?
P values are inconsistently reported as 0.00 or 0.000. us either 2 or 3 decimal places throughout the text. Usually reported as p<0.01 or p<0.001.
Comments on the Quality of English Languagethe ording of some of the sentences could be improved to convey a clear message.
Author Response
Subject is relevant in the context of increasing TAVR numbers worldwide and increasing uptake for intermediate and even low risk patients. NID data was used from 2018-2022. It is not clear why the NRD readmission data was for the years 2021-2022 only (line 84)?
REPLY: In general, the data in NRD is 2-3 times the data in the NIS, as it includes more hospital regions: hence only the latest NRD versions ( 2021 and 2022 ) were used for the readmission analysis. While NIS was used from 2018 - 2022 for general trends, the rationale for using just the latest two versions of the NRD was to get the most updated trends in readmission events
They reported increased TAVR utilization, decreased mortality but high readmissions after TAVR. The data for readmissions was not analysed for TAVR risk group which would have been more meaningful. The manuscript would benefit from some re-organization and better meaningful presentation of data. The presentation of data is unclear and tables have not been adequately utilized for a clearer presentation. Some of the figures are irrelevant and not needed for the data already presented in the text. The results as they are presented are very incoherent and it is difficult to comprehend or summarize from a read. The authors have not written a conclusion statement to their study in the main body of the text. Please get a statistician to review the manuscript before submission.
REPLY: We did do demographics on TAVR readmissions, but no significant disparities were found, hence avoided in the final data presentation. We have rearranged the results, graphs and discussion section
In the abstract, please present data as percentages rather than absolute case numbers.
Reply: added
Please define all abbreviations at first use – STATA, PCS code etc
Reply: STATA is the name of the software. PCS doesnt have specified abbreviation, but we added abbreviation for ICD
Line 22 - …..28,654 patients had 66,100 readmission events… - please quote this readmissions /patient over the 60 day period.
REPLY: changes have been made
Line 22-23 – please give percentages for heart failure, heart block and stroke. REPLY: changes made in the abstract as well as the main text
Tables – give percentages for all numbers as appropriate:
REPLY: changes made in Table 1
Provide p values for group comparisons for the years.
REPLY: we did do the one way anova but adding it would make the table cumbersome. P values for ANOVA has been added to the discussion part (in the discussion of trends)
Predictors of readmission have not been analysed.
REPLY: the predictors for readmissions for HF, stroke and CHB have been analyzed
Line 91 – please describe the APDRG risk severity index and provide reference
REPLY: description added. We do not have a reference quote . The definition was provided with the HCUP database we use
Line 104 – please remove this figure as it does not convey much other than what is presented in results.
REPLY: Done
Table 2 – This refers to socioeconomic data but there is none of that here. It has age, gender and LOS. There is no data on race or income here that would constitute socioeconomic data. Please check the numbers for Total Hospital Charges ($) 72427.6 and 72427.6. They are the same for both index and readmissions.
REPLY: the table has been removed as it does not add much to the results nor worth discussion
Line 117 – 123; present this data as readmissions /patient and give overall percentages for each.
REPLY: percentages for each of the major admission causes among total admissions has been added
Line 135 -138; are the numbers with confidence presented ORs or correlation co-efficients?
REPLY: OR ( has been added, forest plots added)
Please present this as tables for all (stroke, heart failure and heart block)
REPLY: we have added a comparative bar chart instead of table as we felt the manuscript was text heavy and did not more text tables to represent results
Line 133 – says that AF was no significant. This is inconsistent with what is reported in line 136 - Atrial fibrillation: 1.66(1.30 - 2.14) p value 0.000
REPLY: error from our part: changed
Line 144 - ……..prevalence of LBB, RBB and BFB in TAVR admissions that had complete heart block readmissions…… Were the LBB, RBB, BFB at the time of discharge since the readmissions were clearly for CHB. Abbreviations not described
REPLY: We have specified in the text now that these were documented in the index admissions for TAVR. abbreviations added
Line 187 – 189; LOS declined substantially - how much is substantial? Is the change statistically significant?
Reply: have specified in the discussion that data did not have significant variance in the ANOVA ( p values specified)
P values are inconsistently reported as 0.00 or 0.000. us either 2 or 3 decimal places throughout the text. Usually reported as p<0.01 or p<0.001.
REPLY: changes made
Reviewer 3 Report
Comments and Suggestions for Authors
This is an interesting and well-executed study that offers valuable insight into the national trends and short-term outcomes of Transcatheter Aortic Valve Replacement (TAVR) in the United States using large, robust databases (NIS and NRD). The authors effectively demonstrate increasing procedural volume, improved in-hospital outcomes, and identify key predictors for early post-TAVR complications such as stroke, heart failure, and conduction disturbances. The longitudinal scope, statistical modeling, and inclusion of socioeconomic variables strengthen the manuscript. The rising number of TAVR procedures, despite relatively stable admissions for severe aortic stenosis (AS), is particularly notable and raises important questions about evolving patient selection practices.
However, several aspects require clarification or revision:
-Patient Selection and AS Phenotypes:
The manuscript would benefit from a discussion on the types of AS treated (e.g., high-gradient, low-flow low-gradient, paradoxical low-flow, or even moderate AS). The large increase in TAVR procedures suggests a more liberal approach to patient selection, which should be acknowledged, especially in light of the limitations inherent to the dataset (i.e., lack of echocardiographic or hemodynamic data).
-Impact of the COVID-19 Pandemic:
A notable dip in procedures and AS admissions during 2020 is likely attributable to COVID-19. This is a relevant contextual factor and should be briefly addressed.
-Sentinel Embolic Protection Use:
The reporting of Sentinel device usage only in patients who experienced stroke is incomplete. For meaningful interpretation, data should also be presented for those who did not suffer stroke. Without this, the protective effect (or lack thereof) of the device cannot be adequately assessed.
-Hospitalization Costs:
While total hospital charges and trends over time are interesting, this section may be overly detailed given the limited relevance outside the U.S. For an international readership, this part could be streamlined or condensed to focus on key insights rather than granular dollar amounts.
-Clarity in Index vs. Readmission Data:
The distinction between index admissions and readmissions should be clearer, particularly regarding reported hospital stay lengths, costs, and complication rates.
-Statistical Approach:
The rationale for using probit models should be briefly explained, especially since logistic regression is more commonly used in similar analyses.
-Language and Grammar:
The manuscript would benefit from careful proofreading to improve clarity, grammar, and sentence structure, particularly in the introduction and methods sections.
-Figures and Tables:
A side-by-side visualization of Sentinel use and stroke outcomes would improve reader understanding. Data should also be presented for those who did not suffer stroke. It could be included as an additional figure.
-Conclusion
This is a clinically relevant and well-designed study. With revisions focusing on Sentinel data interpretation, patient selection explanation, and streamlining of cost-related content, the manuscript will be significantly enhanced and well-suited for publication.
Author Response
-Patient Selection and AS Phenotypes: Selection criteria with Age > 18 and PCS codes for TAVR mentioned. Unfortunately, the nature of the NIS prevents us from stratifying phenotypes
The manuscript would benefit from a discussion on the types of AS treated (e.g., high-gradient, low-flow low-gradient, paradoxical low-flow, or even moderate AS). The large increase in TAVR procedures suggests a more liberal approach to patient selection, which should be acknowledged, especially in light of the limitations inherent to the dataset (i.e., lack of echocardiographic or hemodynamic data).
Reply: a paragraph in the beginning of the discussion mentioning low flow low gradient versions etc added. The liberal patient selection added with guidelines cited
-Impact of the COVID-19 Pandemic:A notable dip in procedures and AS admissions during 2020 is likely attributable to COVID-19. This is a relevant contextual factor and should be briefly addressed.REPLY: added
-Sentinel Embolic Protection Use:
The reporting of Sentinel device usage only in patients who experienced stroke is incomplete. For meaningful interpretation, data should also be presented for those who did not suffer stroke. Without this, the protective effect (or lack thereof) of the device cannot be adequately assessed.
REPLY: a total of 3456 index admissions for TAVR had documented SENTINEL system: we tried to assess the odd of association of no SENTINEL with stroke: but collinearity error was shown due to the low number of SENTINEL use compared to total TAVR admissions, hence was avoided. 16 patients who had the SENTINEL system had documented strokes: these data have been added
While total hospital charges and trends over time are interesting, this section may be overly detailed given the limited relevance outside the U.S. For an international readership, this part could be streamlined or condensed to focus on key insights rather than granular dollar amounts.
REPLY: True. We have changed the discussion section and took out parts that says signficiant decline. We also added the non significant P value of One way anova showing no significant variance over the years
The distinction between index admissions and readmissions should be clearer, particularly regarding reported hospital stay lengths, costs, and complication rates.
REPLY: in the absence of no signficant disparities among index admissions and readmissions, we have omitted the comparison table.
The rationale for using probit models should be briefly explained, especially since logistic regression is more commonly used in similar analyses.
REPLY: added in the methodology: the only reason we used probit models were because of the large sample size > 50, 000
The manuscript would benefit from careful proofreading to improve clarity, grammar, and sentence structure, particularly in the introduction and methods sections.
REPLY: changes have been made
A side-by-side visualization of Sentinel use and stroke outcomes would improve reader understanding. Data should also be presented for those who did not suffer stroke. It could be included as an additional figure.
REPLY; changes have been made. Figure added
Round 2
Reviewer 2 Report
Comments and Suggestions for Authors
the authors have mad ethe suggested changes and manuscript is much improved and acceptable
Comments on the Quality of English Languagesatisfactory